# Avoiding Pitfalls for Privacy Accounting of Subsampled Mechanisms under Composition

## Abstract

We consider the problem of computing tight privacy guarantees for the composition of subsampled differentially private mechanisms. Recent algorithms can numerically compute the privacy parameters to arbitrary precision but must be carefully applied.

Our main contribution is to address two common points of confusion. First, some privacy accountants assume that the privacy guarantees for the composition of a subsampled mechanism are determined by self-composing the worst-case datasets for the uncomposed mechanism. We show that this is not true in general. Second, Poisson subsampling is sometimes assumed to have similar privacy guarantees compared to sampling without replacement. We show that the privacy guarantees may in fact differ significantly between the two sampling schemes. In particular, we give an example of hyperparameters that result in $\varepsilon \approx 1$ for Poisson subsampling and $\varepsilon > 10$ for sampling without replacement. This occurs for some parameters that could realistically be chosen for DP-SGD.

## 1 Introduction

A fundamental property of differential privacy is that the composition of multiple differentially private mechanisms still satisfies differential privacy. This property allows us to design complicated mechanisms with strong formal privacy guarantees such as differentially private stochastic gradient descent (DP-SGD, [SCS13, BST14, ACG$^+$16]).

The privacy guarantees of a mechanism inevitably deteriorate with the number of compositions. Accurately quantifying the privacy parameters under composition is highly non-trivial and is an important area within the field of differential privacy. A common approach is to find the privacy parameters for each part of a mechanism and apply a composition theorem [DRV10, KOV15] to find the privacy parameters of the full mechanism. In recent years, several alternatives to the traditional definition of differential privacy with cleaner results for composition have gained popularity (see, e.g., [DR16, BS16, Mir17, DRS19]).

Another important concept is privacy amplification by subsampling (see, e.g., [BBG18, Ste22]). The general idea is to improve privacy guarantees by only using a randomly sampled subset of the full dataset as input to a mechanism. In this work we consider the problem of computing tight privacy parameters for subsampled mechanisms under composition.

One of the primary motivations for studying privacy accounting of subsampled mechanisms is DP-SGD. DP-SGD achieves privacy by clipping gradients and adding Gaussian noise to each batch. As such, we can find the privacy parameters by analyzing the subsampled Gaussian mechanism under composition. One of the key contributions of [ACG$^+$16] was the moments accountant, which gives tighter bounds for the mechanism than the generic composition theorems. Later work

improved the accountant by giving improved bounds on the Rényi Differential Privacy guarantees of the subsampled Gaussian mechanism under both Poisson subsampling and sampling without replacement [MTZ19, WBK20].

Even small constant factors in an $(\varepsilon, \delta)$-DP budget are important. First, from the definition, such constant factors manifest exponentially in the privacy guarantee. Furthermore, when training a model privately with DP-SGD, it has been observed that they can lead to significant differences in the downstream utility, see, e.g., Figure 1 of [DBH+22]. Consequently, "saving" such a factor in the value of $\varepsilon$ through tighter analysis can be very valuable. While earlier *approximate* techniques for privacy accounting (e.g., moments accountant of [ACG+16] and related methods) were lossy, a more recent line of work focuses on *exact* computation of privacy loss by numerically estimating the privacy parameters [SMM19, KJH20, KJPH21, GLW21, ZDW22]. These accountants generally look at the "worst case" for a single iteration for a privacy mechanism, and then use a fast Fourier transform (FFT) to compose the privacy loss over multiple iterations. They often rely on an implicit assumption that the worst-case dataset for a single execution of a privacy mechanism remains the worst case for a self-composition of the mechanism.

Most privacy accounting techniques for DP-SGD assume a version of the algorithm that employs amplification by *Poisson* subsampling. That is, the batch for each iteration is formed by including each point independently with sampling probability $\gamma$. Other privacy accountants consider a variant where random batches of a fixed size are selected for each step. Note that both of these are inconsistent with the standard method in the non-private setting, where batches are formed by randomly permuting and then partitioning the dataset. Indeed, the latter approach is much more efficient, and highly-optimized in most libraries. Consequently, many works in private machine learning implement a method with the conventional shuffle-and-partition method of batch formation, but employ privacy accountants that assume some other method of sampling batches. The hope is that small modifications of this sort would have negligible impact on the privacy analysis, thus justifying privacy accountants for a setting which is *technically* not matching. Concurrent work to this paper by [CGK+24] compares the shuffle-and-partition technique with Poisson subsampling. Similar to our results they find that the batching method can significantly impact the privacy parameters.

The central aim of our paper is to highlight and clarify some common problems with privacy accounting techniques. Towards the goal of more faithful comparisons between private algorithms that rely upon such accountants, we make the following contributions:

- In Sections 4 and 5, we establish that a worst-case dataset may exist for a single execution of a privacy mechanism but may fail to exist when looking at the self-composition of the same mechanism. Some popular privacy accountants incorrectly assume otherwise. Our counterexample involves the subsampled Laplace mechanism, and stronger analysis is needed to demonstrate the soundness of privacy accountants for specific mechanisms, e.g., the subsampled Gaussian mechanism.

- In Section 6, we show that rigorous privacy accounting is *significantly* affected by the method of sampling batches, e.g., Poisson versus fixed-size. This results in sizeable differences in the resulting privacy guarantees for settings which were previously treated as interchangeable by prior works. Consequently, we caution against the common practice of using one method of batch sampling and employing the privacy accountant for another.

- In Section 7, we discuss issues that arise in tight privacy accounting under the "substitution" relation for neighbouring datasets, which make this setting even more challenging than under the traditional "add/remove" relation. Once again we consider the subsampled Laplace mechanism and show that there may be several worst-case datasets one must consider when doing accounting, exposing another important gap in existing analyses.

## 2 Preliminaries

Differential privacy is a rigorous privacy framework introduced by [DMNS06]. Differential privacy is a restriction on how much the output distribution of a mechanism can change between any pair of datasets that differ only in a single individual. Such datasets are called neighboring, and we denote a pair of neighboring datasets as $D \sim D'$. We formally define neighboring datasets below.

**Definition 1** (($\varepsilon, \delta$)-Differential Privacy). *A randomized mechanism $\mathcal{M}$ satisfies ($\varepsilon, \delta$)-DP under neighboring relation $\sim$ if and only if for all $D \sim D'$ and all measurable sets of outputs $Z$ we have*

$$\Pr[\mathcal{M}(D) \in Z] \le e^\varepsilon \Pr[\mathcal{M}(D') \in Z] + \delta.$$

In this work, we consider problems where we want to estimate a sum for $k$ queries where each datapoint holds a single-dimensional real value in the interval $[-1, 1]$ for each query. The mechanisms we consider apply more generally to multi-dimensional real-valued queries. Since we demonstrate issues already present in the former more restrictive setting, these pitfalls are present in the more general case as well. We focus on single-dimensional inputs for simplicity of presentation. Likewise, by considering mechanisms defined on $[-1, 1]$, our privacy analysis immediately extends to any mechanism defined on $\mathbb{R}$ that clips to $[-1, 1]$. After the appropriate rescaling, our privacy analysis extends to any mechanism used in practice for DP-SGD. Note that in all but one example in Section 7 the datapoints hold the same value for all $k$ queries for the datasets we consider. We abuse notation and represent each data point as a single real value rather than a vector.

On the domain $[-1, 1]^{* \times k} := \bigcup_{m=0}^{\infty} [-1, 1]^{m \times k}$, we define the neighboring definitions of add, remove, and substitution (replacement). We typically want the neighboring relation to be symmetric, which is why add and remove are typically included in a single definition. However, as noted by previous work we need to analyze the add and remove cases separately to get tight results (see, e.g., [ZDW22]).

**Definition 2** (Neighboring Datasets). *Let $D$ and $D'$ be datasets. If $D'$ can be obtained by adding a datapoint to $D$, then we write $D \sim_A D'$. Likewise, if $D'$ can be obtained by removing a datapoint from $D$, then we write $D \sim_R D'$. Combining these, write $D \sim_{A/R} D'$ if $D \sim_A D'$ or $D \sim_R D'$. Finally, we write $D \sim_S D'$ if $D$ can be obtained from $D'$ by swapping one datapoint for another.*

Note that differential privacy under add and remove implies differential privacy under substitution, with appropriate translation of the privacy parameters.

Definition 1 can be restated in terms of the hockey-stick divergence.

**Definition 3** (Hockey-stick Divergence). *For any $\alpha \ge 0$ the hockey-stick divergence between two distributions $P$ and $Q$ is defined as*

$$H_\alpha(P||Q) := \mathbb{E}_{y \sim Q}\left[\max\left\{\frac{dP}{dQ}(y) - \alpha, 0\right\}\right]$$

*where $\frac{dP}{dQ}$ is the Radon–Nikodym derivative.*

Specifically, a randomized mechanism $\mathcal{M}$ satisfies ($\varepsilon, \delta$)-DP if and only if $H_{e^\varepsilon}(\mathcal{M}(D)||\mathcal{M}(D')) \le \delta$ for all pairs of neighboring datasets $D \sim D'$. This restated definition is the basis for the privacy accounting tools we consider in this paper. If we know what choice of neighboring datasets $D \sim D'$ maximizes the expression then we can get optimal parameters by computing $H_{e^\varepsilon}(\mathcal{M}(D)||\mathcal{M}(D'))$.

The full range of privacy guarantees for a mechanism can be captured by the privacy curve.

**Definition 4** (Privacy Curves). *The privacy curve of a randomized mechanism $\mathcal{M}$ under neighboring relation $\sim$ is the function $\delta_{\mathcal{M}}^{\sim} : \mathbb{R} \to [0, 1]$ given by*

$$\delta_{\mathcal{M}}^{\sim}(\varepsilon) := \min\{\delta \in [0, 1] : \mathcal{M} \text{ is } (\varepsilon, \delta)\text{-DP}\}.$$

*If there is a single pair of neighboring datasets $D \sim D'$ such that $\delta_{\mathcal{M}}^{\sim}(\varepsilon) = H_{e^\varepsilon}(\mathcal{M}(D)||\mathcal{M}(D'))$ for all $\varepsilon \ge 0$, we say that the privacy curve of $\mathcal{M}$ under $\sim$ is realized by the worst-case dataset pair $(D, D')$.*

Unfortunately, a worst-case dataset pair does not always exist. A broader tool that is now frequently used in the computation of privacy curves is the privacy loss distribution (PLD) formalism [DR16, SMM19].

**Definition 5** (Privacy Loss Distribution). *Given a mechanism $\mathcal{M}$ and a pair of neighboring datasets $D \sim D'$, the privacy loss distribution of $\mathcal{M}$ with respect to $(D, D')$ is*

$$L_{\mathcal{M}}(D||D') := \ln(d\mathcal{M}(D)/d\mathcal{M}(D'))(y),$$

*where $y \sim \mathcal{M}(D)$ and $d\mathcal{M}(D)/d\mathcal{M}(D')$ means the density of $\mathcal{M}(D)$ with respect to $\mathcal{M}(D')$.*

An important caveat is that the privacy loss distribution is defined with respect to a specific pair of datasets, whereas the privacy curve implicitly involves taking a maximum over all neighboring pairs of datasets. Nonetheless, the PLD formalism can be used to recover the hockey-stick divergence via

$$H_{e^\varepsilon}(\mathcal{M}(D)||\mathcal{M}(D')) = \mathbb{E}_{Y \sim L_{\mathcal{M}(D||D')}}[1 - e^{\varepsilon - Y}],$$

from which we can reconstruct the privacy curve as

$$\widetilde{\delta}_{\mathcal{M}}(\varepsilon) = \max_{D \sim D'} \mathbb{E}_{Y \sim L_{\mathcal{M}(D||D')}}[1 - e^{\varepsilon - Y}].$$

Lastly, we define the two subsampling procedures we consider in this work: sampling without replacement (WOR) and Poisson sampling. Given a dataset $D = (x_1, \ldots, x_n)$ and a set $I \subseteq \{1, \ldots, n\}$, we denote the restriction of $D$ to $I = \{i_1, \ldots, i_b\}$ by $D|_I := (x_{i_1}, \ldots, x_{i_b})$.

**Definition 6** (Subsampling). *Let $\mathcal{M}$ take datasets of size[1] $b \geq 1$. The $\binom{n}{b}$-subsampled mechanism $\mathcal{M}_{WOR}$ is defined on datasets of size $n \geq b$ as*

$$\mathcal{M}_{WOR}(D) := \mathcal{M}(D|_I),$$

*where $I$ is a uniform random $b$-subset of $\{1, \ldots, n\}$.*

*On the other hand, given a mechanism $\mathcal{M}$ taking datasets of any size, the $\gamma$-subsampled mechanism $\mathcal{M}_{Poisson}$ is defined on datasets of arbitrary size as*

$$\mathcal{M}_{Poisson}(D) := \mathcal{M}(D|_I),$$

*where $I$ includes each element of $\{1, \ldots, |D|\}$ independently with probability $\gamma$.*

# 3 Related Work

After [DR16] introduced privacy loss distributions, a number of works used the formalism to estimate the privacy curve to arbitrary precision, beginning with [SMM19]. [KJH20, KJPH21] developed an efficient accountant that efficiently computes the convolution of PLDs by leveraging the fast Fourier transform. [GLW21] fine-tuned the application of FFT to speed up the accountant by several orders of magnitude.

The most relevant related paper for our work is by [ZDW22]. They introduce the concept of a dominating pair of distributions. Dominating pairs generalize worst-case datasets, which for some problems can be difficult to find and may not even exist.

**Definition 7** (Dominating Pair of Distributions [ZDW22]). *The ordered pair $(P, Q)$ is a dominating pair of distributions for a mechanism $\mathcal{M}$ (under some neighboring relation $\sim$) if for all $\alpha \geq 0$ it holds that*

$$\sup_{D \sim D'} H_\alpha(\mathcal{M}(D)||\mathcal{M}(D')) \leq H_\alpha(P||Q).$$

The hockey-stick divergence of the dominating pair $P$ and $Q$ gives an upper bound on the value $\delta$ for any $\varepsilon$. Note that the distributions $P$ and $Q$ do not need to be output distributions of the mechanism. However, if there exists a pair of neighboring datasets such that $P = \mathcal{M}(D)$ and $Q = \mathcal{M}(D')$ then we can find tight privacy parameters by analyzing the mechanisms with inputs $D$ and $D'$ because $H_{e^\varepsilon}(\mathcal{M}(D)||\mathcal{M}(D'))$ is also a lower bound on $\delta$ for any $\varepsilon$. We refer to such $D \sim D'$ as a dominating pair of datasets.

The definition of dominating pairs of distributions is useful for analyzing the privacy guarantees of composed mechanisms. In this work, we focus on the special case where a mechanism consists of $k$ self-compositions. This is, for example, the case in DP-SGD, in which we run several iterations of the subsampled Gaussian mechanism. The property we need for composition is presented in Theorem 8.

**Theorem 8** (Following Theorem 10 of [ZDW22]). *If $(P, Q)$ is a dominating pair for a mechanism $\mathcal{M}$ then $(P^k, Q^k)$ is a dominating pair for $k$ iterations of $\mathcal{M}$.*

When studying differential privacy parameters in terms of the hockey-stick divergence, we usually focus on the case of $\alpha \geq 1$. Recall that the hockey-stick divergence of order $\alpha$ can be used to bound

---

[1]We treat the sample size and batch size as public knowledge in line with prior work [ZDW22].

the value of $\delta$ for an $(\varepsilon, \delta)$-DP mechanism where $\varepsilon = \ln(\alpha)$. We typically do not care about the region of $\alpha < 1$ because it corresponds to negative values of $\varepsilon$. However, the definition of dominating pairs of distributions must include these values as well. This is because outputs with negative privacy loss are important for composition and Theorem 8 would not hold if the definition only considered $\alpha \geq 1$. In Sections 5 and 7 we consider mechanisms where the distributions that bound the hockey-stick divergence for $\alpha \geq 1$ without composition do not bound the divergence for $\alpha \geq 1$ under composition.

[ZDW22] studied general mechanisms in terms of dominating pairs of distributions under Poisson subsampling and sampling without replacement. Their work gives upper bounds on the privacy parameters based on the dominating pair of distributions of the non-subsampled mechanism. We use some of their results which we introduce later throughout this paper.

# 4 Dominating Pair of Datasets under Add and Remove Relations

In this section we give pairs of neighboring datasets with provable worst-case privacy parameters under the add and remove neighboring relations separately. We use these datasets as examples of the pitfalls to avoid in the subsequent section, where we discuss the combined add/remove neighboring relation.

**Proposition 9.** *Let $\mathcal{M}$ be either the Gaussian mechanism $\mathcal{M}(x_1, \ldots, x_n) := \sum_{i=1}^{n} x_i + \mathcal{N}(0, \sigma^2)$ or the Laplace mechanism $\mathcal{M}(x_1, \ldots, x_n) := \sum_{i=1}^{n} x_i + \mathrm{Lap}(0, s)$.*

> *1. The datasets $D := (0, \ldots, 0)$ and $D' := (0, \ldots, 0, 1)$ form a dominating pair of datasets for $\mathcal{M}_{Poisson}$ under the add relation and $(D', D)$ is a dominating pair of datasets under the remove relation.*

> *2. Likewise, the datasets $D := (-1, \ldots, -1)$ and $D' := (-1, \ldots, -1, 1)$ form a dominating pair of datasets for $\mathcal{M}_{WOR}$ under the add relation and $(D', D)$ is a dominating pair of datasets under the remove relation.*

The proposition implies that the hockey-stick divergence of the mechanisms with said datasets as input describes the privacy curves of the composed mechanisms under the add and remove relations, respectively. We contrast this good behavior of composed and subsampled mechanisms under add and remove separately with the Laplace mechanism, which, as we will see in Section 5, does not behave well when composed under the combined add/remove relation.

Our dominating pair of datasets can be found by reduction to one of the main results of [ZDW22].

**Theorem 10** (Theorem 11 of [ZDW22]). *Let $\mathcal{M}$ be a randomized mechanism, let $\mathcal{M}_{Poisson}$ be the $\gamma$-subsampled version of the mechanism, and let $\mathcal{M}_{WOR}$ be the $\binom{n}{b}$-subsampled version of the mechanism on datasets of size $n$ and $n - 1$ with $\gamma = b/n$.*

> *1. If $(P, Q)$ dominates $\mathcal{M}$ for add neighbors then $(P, (1 - \gamma)P + \gamma Q)$ dominates $\mathcal{M}_{Poisson}$ for add neighbors and $((1 - \gamma)Q + \gamma P, P)$ dominates $\mathcal{M}_{Poisson}$ for removal neighbors.*

> *2. If $(P, Q)$ dominates $\mathcal{M}$ for substitution neighbors then $(P, (1 - \gamma)P + \gamma Q)$ dominates $\mathcal{M}_{WOR}$ for add neighbors and $((1 - \gamma)P + \gamma Q, P)$ dominates $\mathcal{M}_{WOR}$ for removal neighbors.*

In Appendix A we prove that Proposition 9 holds by showing that the hockey-stick divergence between the mechanism with the dominating pairs of datasets matches the upper bound from Theorem 10.

Crucially, Proposition 9 implies that under the add and remove relations, we must add noise with twice the magnitude when sampling without replacement compared to Poisson subsampling! The intuition behind this difference is that the subroutine behaves similarly to the add/remove neighboring relation when using Poisson subsampling, whereas it resembles the substitution neighborhood when sampling without replacement. When $D'_i$ is included in the batch another datapoint is 'pushed out' of the batch under sampling without replacement. Due to this parallel one might hope that the difference in privacy parameters between Poisson subsampling and sampling without replacement only differ by a small constant similar to the difference between the add/remove and substitution neighboring relations. That is indeed the case for many parameters, but as we show in Section 7 this assumption unfortunately does not always hold.

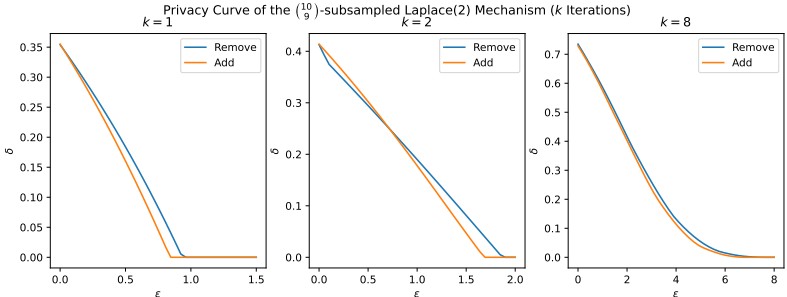

Figure 1: The privacy curves for the subsampled Laplace mechanism under the remove and add neighboring relations respectively.

## 5 No Worst-case Pair of Datasets under Add/Remove Relation

So far, we have considered the entire privacy curve for all $\varepsilon \in \mathbb{R}$. This is a necessary subtlety for PLD privacy accounting tools under composition (e.g., Theorem 8). Here we focus only on the privacy curve for $\varepsilon \geq 0$. Our main result of this section is to give a minimal example of a mechanism $\mathcal{M}$ that admits a worst-case dataset pair under $\sim_{A/R}$ yet $\mathcal{M}^k$ does not admit any worst-case dataset pair for some $k > 1$. This violates an implicit assumption made by some privacy accountants.

**Proposition 11.** *For some mechanism $\mathcal{M}$, the privacy curve of the $\binom{n}{b}$-subsampled mechanism $\mathcal{M}_{WOR}$ is realized by a pair of datasets under $\sim_{A/R}$, yet no pair of datasets realizes the privacy curve of $\mathcal{M}_{WOR}^k$ for all $k > 1$.*

A proof of this proposition for a simple mechanism can be found in Appendix B.1. However, it is more illustrative to demonstrate the proposition informally for the Laplace mechanism $\mathcal{M}$. In this case, note that the proposition can be extended to $\mathcal{M}_{Poisson}$ as well. The proposition stands in contrast to the case of the add and remove relations discussed in Proposition 9. That is, we can find datasets $D \sim_A D'$ such that $\delta_{\mathcal{M}_{WOR}}^{\sim A}$ is realized by $(D, D')$ and $\delta_{\mathcal{M}_{WOR}}^{\sim R}$ is realized by $(D', D)$, but no such (ordered) pair realizes the privacy curve under $\sim_{A/R}$.

Moreover, it is generally the case that the privacy curve of a subsampled mechanism without composition under $\sim_R$ dominates the privacy curve under $\sim_A$ when $\varepsilon \geq 0$ (see, e.g., Proposition 30 of [ZDW22] or Theorem 5 of [MTZ19]). Specifically, it follows from Proposition 30 of [ZDW22] that in the case of the subsampled Laplace mechanism and $\varepsilon \geq 0$, we have that

$$\delta_{\mathcal{M}_{WOR}}^{\sim A/R}(\varepsilon) = \delta_{\mathcal{M}_{WOR}}^{\sim R}(\varepsilon) \geq \delta_{\mathcal{M}_{WOR}}^{\sim A}(\varepsilon).$$

Here we visualize the counter-example by plotting privacy curves for the add and remove relation in Figure 1. Note that $\delta_{\mathcal{M}_{WOR}}^{\sim A/R}(\varepsilon) = \max\{\delta_{\mathcal{M}_{WOR}}^{\sim A}(\varepsilon), \delta_{\mathcal{M}_{WOR}}^{\sim R}(\varepsilon)\}$. Figure 1 shows several variations of the curves $\delta_{\mathcal{M}_{WOR}^k}^{\sim A}$ and $\delta_{\mathcal{M}_{WOR}^k}^{\sim R}$, which we estimated numerically by Monte Carlo simulation (as in, e.g., [WMW+23]). Appendix B.2 has the methodological details. These curves are seen to cross in the region $\varepsilon \geq 0$ for $k = 2$ compositions.

The phenomenon is most apparent for $k = 2$. There is a clear break in the curve for the remove relation. Under many compositions, however, it is known that both PLDs converge to a Gaussian distribution [DRS19], which explains why this break vanishes as the number of compositions increases.

**Avoiding incorrect upper bounds** As shown in this section we cannot assume that the privacy curve for the remove relation dominates the add relation for composed subsampled mechanisms under $\sim_{A/R}$ even though it is the case without composition. Luckily, this particular issue can be easily resolved by computing the privacy parameters for the add and remove relation separately and taking the maximum. This technique is already used in practice in, e.g., the Google DP library [Goo20].

We conjecture that this workaround is unnecessary for the Gaussian mechanism—the natural choice for DP-SGD. We searched a wide range of parameters and were unable to produce a counterexample.

**Conjecture 12.** *Let $\mathcal{M}$ be the Gaussian mechanism with any $\sigma$. Then for all $k > 0$, $\gamma \in [0, 1]$, and $\varepsilon \geq 0$ we have*

$$\delta_{\mathcal{M}_{Poisson}^k}^{\sim A/R}(\varepsilon) = \delta_{\mathcal{M}_{Poisson}^k}^{\sim R}(\varepsilon) \geq \delta_{\mathcal{M}_{Poisson}^k}^{\sim A}(\varepsilon).$$

## 6 Comparison of Sampling Schemes

In this section we explore the difference in privacy parameters between Poisson subsampling and sampling without replacement. We focus on the subsampled Gaussian mechanism which is the mechanism of choice for DP-SGD. We show that for some parameters the privacy guarantees of the mechanism differ significantly between the two sampling schemes.

There are several different techniques one might use when selecting privacy-specific hyperparameters for DP-SGD. One approach is to fix the value of $\delta$ and the number of iterations. Given a sampling rate $\gamma$ and a value for $\varepsilon$, we can compute the smallest value for the noise multiplier $\sigma$ such that the mechanism satisfies $(\varepsilon, \delta)$-differential privacy. We use this approach to showcase our findings. We fix $\delta = 10^{-6}$ and the number of iterations to $10,000$. We then vary the sampling rate between $10^{-4}$ to 1 and use the *PLD* accountant implemented in the Opacus library [YSS+21] to compute $\sigma$.

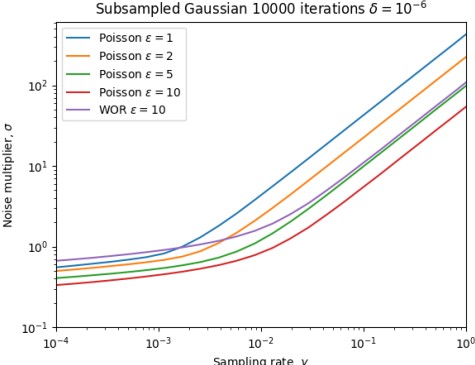

Figure 2: Plots of the smallest noise multiplier $\sigma$ required to achieve certain privacy parameters for the subsampled Gaussian mechanism with varying sampling rates under add/remove. Each line shows a specific value of $\varepsilon$ for either Poisson subsampling or sampling without replacement. The parameter $\delta$ is fixed to $10^{-6}$ for all lines.

In Figure 2 we plot the noise multiplier required to achieve $(\varepsilon, 10^{-6})$-DP with Poisson subsampling for $\varepsilon \in \{1, 2, 5, 10\}$. For comparison, we plot the noise multiplier that achieves $(10, 10^{-6})$-DP when sampling without replacement. Recall from Section 4 that the noise magnitude required when sampling without replacement is exactly twice that required for Poisson subsampling. The plots are clearly divided into two regions. For large sampling rate, the noise multiplier scales roughly linearly in the sampling rate. However, for sufficiently low sampling rates the noise multiplier decreases much slower. This effect has been observed previously for setting hyperparameters (see Figure 1 of [PHK+23] for a similar plot).

| $\delta$ | $\varepsilon$ (Poisson) | $\varepsilon$ (WOR) |
|---|---|---|
| $10^{-7}$ | 1.19 | 17.48 |
| $10^{-6}$ | 0.96 | 15.26 |
| $10^{-5}$ | 0.80 | 12.98 |
| $10^{-4}$ | 0.64 | 10.62 |

Table 1: The table contrasts the privacy parameter $\varepsilon$ for the subsampled Gaussian mechanism with $10,000$ iterations, sampling rate $\gamma = 0.001$, and noise multiplier $\sigma = 0.8$ for multiple values of $\delta$.

**Avoiding problematic parameters** It is generally advised to select parameters that fall into the right-hand regime of the plots in Figure 2 [PHK+23]. However, one might select parameters close to the transition point. This can be especially problematic if the wrong privacy accountant is used. The transition point happens when $\sigma$ is slightly less than 1 for Poisson sampling and therefore it happens when it is slightly less than 2 for sampling without replacement. The consequence can be seen for the plot for sampling without replacement in Figure 2. When the sampling rates are high the noise required roughly matches that for $\varepsilon = 5$ with Poisson subsampling. But when the sampling rate is small we have to add more noise than is required for $\varepsilon = 1$ with Poisson subsampling. As such, if we

use a privacy accountant for Poisson subsampling and have a target of $\varepsilon = 1$ but our implementation uses sampling without replacement the actual value of $\varepsilon$ could be above 10! We might hope that this increase would be offset if we allow for some slack in $\delta$ as well. However, as seen in the table of Figure 1 there can still be a big gap in $\varepsilon$ between the sampling schemes even when we allow a difference of several orders of magnitude in $\delta$.

# 7 Substitution Neighboring Relation

In this section, we consider both sampling schemes under the substitution neighboring relation. In their work on computing tight differential privacy guarantees, [KJH20] considered worst-case distributions for the subsampled Gaussian mechanism under multiple sampling techniques and neighboring relations. In the substitution case, they compute the hockey-stick divergence between $(1 - \gamma)\mathcal{N}(0, \sigma^2) + \gamma\mathcal{N}(-1, \sigma^2)$ and $(1 - \gamma)\mathcal{N}(0, \sigma^2) + \gamma\mathcal{N}(1, \sigma^2)$. These distributions correspond to running the mechanism with neighboring datasets where all but one entry is 0. We first consider Poisson subsampling in the proposition below and later discuss sampling without replacement.

**Proposition 13.** *Consider the Gaussian mechanism $\mathcal{M}(x_1, \ldots, x_n) := \sum_{i=1}^{n} x_i + \mathcal{N}(0, \sigma^2)$ and let $\mathcal{M}_{Poisson}$ be the $\gamma$-subsampled mechanism. Then $D := (0, \ldots, 0, 1)$ and $D' := (0, \ldots, 0, -1)$ form a dominating pair of datasets under the substitution neighboring relation.*

Proposition 13 simply confirms that the pair of distributions considered by [KJH20] does indeed give correct guarantees as it is a dominating pair of distributions. However, as far as we are aware, no formal proof existed anywhere. Our proof of the proposition is in Appendix C.

In the rest of the section we focus on sampling without replacement. We start by restating another result from [ZDW22] which we use throughout the section.

**Theorem 14** (Proposition 30 of [ZDW22])**.** *If $(P, Q)$ dominates $\mathcal{M}$ under substitution for datasets of size $\gamma n$, then under the substitution neighborhood for datasets of size $n$, we have*

$$\delta(\alpha) \leq \begin{cases} H_\alpha((1 - \gamma)Q + \gamma P || P) & \text{if } \alpha \geq 1; \\ H_\alpha(P || (1 - \gamma)P + \gamma Q) & \text{if } 0 < \alpha < 1, \end{cases}$$

*where $\delta(\alpha)$ is the largest hockey-stick divergence of order $\alpha$ for $\mathcal{M}_{WOR}$ on neighboring datasets.*

Next, we address a mistake made in related work. We introduced the distributions considered by [KJH20] for Poisson subsampling above and we show in Proposition 13 that it is a dominating pair of distributions. However, [KJH20] claimed in their paper that the privacy curves are identical for the two sampling schemes under the substitution relation which is unfortunately incorrect.

They considered datasets where all but one entry has a value of 0. This results in correct distributions for Poisson subsampling but for sampling without replacement, we instead consider the datasets $D := (-1, \ldots, -1, 1)$ and $D' := (-1, \ldots, -1, -1)$. With these datasets the values of $H_\alpha(\mathcal{M}_{WOR}(D) || \mathcal{M}_{WOR}(D'))$ and $H_\alpha(\mathcal{M}_{WOR}(D') || \mathcal{M}_{WOR}(D))$ match the cases of the upper bound in Theorem 14 for $\alpha \geq 1$ and $\alpha < 1$, respectively. This can be easily verified by following the steps of the proof of Proposition 9 for sampling without replacement.

We can use the datasets above to compute tight privacy guarantees for a single iteration. However, composition is more complicated since neither of the two directions corresponds to a dominating pair of distributions. One might hope that we could simply compute the hockey-stick divergence of the self-composed distributions in both directions and use the maximum similar to the add/remove case. However, for some mechanisms that is not sufficient because we can combine the directions unlike with the add and remove cases. Next we give a minimal counterexample using the Laplace mechanism to showcase this challenge.

We consider datasets of size 2 and sample batches with a single element such that $\gamma = 0.5$. Let $x_1$ and $x_2$ denote the two data points in $D$ and without loss of generality assume that $x_1 = x'_1$ and $x_2 \neq x'_2$, where $x'_1$ and $x'_2$ are the corresponding data points in $D'$. We apply the subsampled Laplace mechanism with a scale of 2 and perform 2 queries where $x_1$ has the value $-1$ for both queries. Let $P := 0.5 \cdot \text{Lap}(-1, 2) + 0.5 \cdot \text{Lap}(1, 2)$ and $Q := \text{Lap}(-1, 2)$. That is, $P$ and $Q$ are the distributions for running one query of $\mathcal{M}_{WOR}(D)$ with $x_2$ having value 1 or $-1$, respectively. Then $H_{e^\varepsilon}(P \times P || Q \times Q)$ is the hockey-stick divergence for the mechanism if $x_2$ has value 1 for

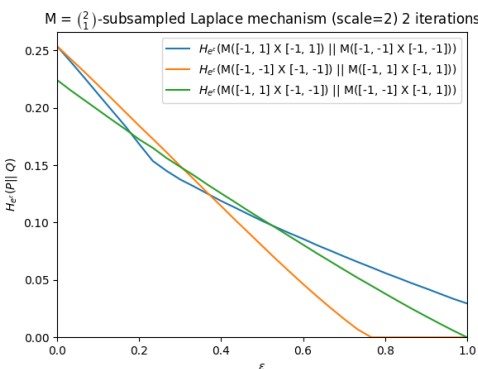

Figure 3: Hockey-stick divergence of the Laplace mechanism when sampling without replacement under $\sim_S$. The worst-case pair of datasets depends on the value of $\varepsilon$.

both queries and $x_2'$ has value $-1$ for both queries. Similarly, $H_{e^\varepsilon}(Q \times Q || P \times P)$ is the divergence when $x_2$ has value $-1$ for both queries and $x_2'$ has value $1$ for both queries.

The two hockey-stick divergences above are similar to those for the remove and add neighboring relations. However, we also have to consider $H_{e^\varepsilon}(P \times Q || P \times Q)$ in the case of substitution. These distributions correspond to the case when $x_2$ has a value of $1$ for the first query and $-1$ for the second query, and $x_2'$ has a value of $-1$ for the first query and $1$ for the second query. Figure 3 shows the hockey-stick divergence as a function of $\varepsilon$ for the three pairs of neighboring datasets. The largest divergence depends on the value of $\varepsilon$ with all three divergences being the maximum for some interval. This counterexample shows that we cannot upper bound the hockey-stick divergence for the subsampled Laplace mechanism as $\max\{H_{e^\varepsilon}(P^k || Q^k), H_{e^\varepsilon}(Q^k || P^k)\}$ for $k > 1$. For $k$ compositions, we have to consider $k + 1$ ways of combining $P$ and $Q$. This significantly slows down the accountants in contrast to the 2 cases required for add/remove. Worse still, we do not have a proof that one of $k + 1$ cases is the worst-case pair of datasets for all $\varepsilon \geq 0$.

In Appendix D we use an alternative technique for bounding the privacy curve under the substitution relation based on [DGK+22]. We show that this accountant does not generally outperform the RDP accountant. This demonstrates the need to strengthen the theory for sampling without replacement under the substitution relation for the purposes of tight privacy accounting.

# 8 Discussion

We have highlighted two issues that arise in the practice of privacy accounting.

First, we have given a concrete example where the worst-case dataset (for $\varepsilon \geq 0$) of a subsampled mechanism fails to be a worst-case dataset once that mechanism is composed. Care should therefore be taken to ensure that the privacy accountant computes privacy guarantees with respect to a true worst-case dataset for a given choice of $\varepsilon$.

Secondly, we have shown that the privacy parameters for a subsampled and composed mechanism can differ significantly for different subsampling schemes. This can be problematic if the privacy accountant is assuming a different subsampling procedure from the one actually employed. We have shown this in the case of Poisson sampling and sampling without replacement but the phenomenon is likely to occur when comparing Poisson sampling to shuffling as well. Computing tight privacy guarantees for the shuffled Gaussian mechanism remains an important open problem. It is best practice to ensure that the implemented subsampling method matches the accounting method. When this is not practical, the discrepancy should be disclosed.

We conclude with two recommendations for practitioners applying privacy accounting in the DP-SGD setting. We recommend disclosing the privacy accounting hyperparameters for the sake of reproducibility (see Section 5.3.3 of [PHK+23] for a list of suggestions). Finally, we also recommend that, when comparisons are made between DP-SGD mechanisms, the privacy accounting for both should be re-run for the sake of fairness.

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

 # A  Proof of Proposition 9

Without loss of generality, we show both parts for the Gaussian mechanism under the add neighboring relation only.

We first note that any pair of neighboring datasets with maximum $\ell_2$-distance is a dominating pair of datasets for the Gaussian mechanism [BW18]. Since the datapoints in our setting are from $[-1, 1]$ this implies that $(\mathcal{N}(0, \sigma^2), \mathcal{N}(1, \sigma^2))$ is a dominating pair of distributions for $\mathcal{M}$ under $\sim_A$ and $(\mathcal{N}(r, \sigma^2), \mathcal{N}(r + 2, \sigma^2))$ is a dominating pair of distributions for $\mathcal{M}$ under $\sim_S$ for any $r \in \mathbb{R}$. The distance of 2 is obtained by substituting $-1$ with $1$.

Now, let us prove part 1 of the proposition. To that end, let $D$ be the all zeros dataset and let $D'$ be $D$ with a 1 appended to the end. The sum of the subsampled dataset is 1 if the last datapoint is included in the sample and 0 otherwise. As such, we have that

$$\mathcal{M}_{Poisson}(D') = (1 - \gamma)\mathcal{N}(0, \sigma^2) + \gamma\mathcal{N}(1, \sigma^2)$$

Since $(\mathcal{N}(0, \sigma^2), \mathcal{N}(1, \sigma^2))$ is a dominating pair of distributions for $\mathcal{M}$ under $\sim_A$ from Theorem 10 we have that

$$(\mathcal{N}(0, \sigma^2), (1 - \gamma)\mathcal{N}(0, \sigma^2) + \gamma\mathcal{N}(1, \sigma^2)) = (\mathcal{M}_{Poisson}(D), \mathcal{M}_{Poisson}(D'))$$

dominates $\mathcal{M}_{Poisson}$ under $\sim_A$.

As for part 2, let $\gamma := b/n$ for convenience, let $D$ be the all $-1$ dataset, let $D'$ be $D$ with a single $-1$ substituted for a 1. We can describe $\mathcal{M}_{WOR}(D')$ by considering the two cases where the 1 is either excluded or included in the batch of size $b$

$$\mathcal{M}_{WOR}(D') = (1-\gamma)\mathcal{M}(\underbrace{-1, \ldots, -1, -1}_{b}) + \gamma\mathcal{M}(\underbrace{-1, \ldots, -1, 1}_{b}) = (1-\gamma)\mathcal{N}(-b, \sigma^2) + \gamma\mathcal{N}(-b+2, \sigma^2)$$

Since $(\mathcal{N}(-b, \sigma^2), \mathcal{N}(-b + 2, \sigma^2))$ is a dominating pair of distributions for $\mathcal{M}$ under $\sim_S$ from Theorem 10 we have that

$$(\mathcal{N}(-b, \sigma^2), (1 - \gamma)\mathcal{N}(-b, \sigma^2) + \gamma\mathcal{N}(-b + 2, \sigma^2)) = (\mathcal{M}_{WOR}(D), \mathcal{M}_{WOR}(D'))$$

dominates $\mathcal{M}_{WOR}$ under $\sim_A$.

The proof for the remove direction is symmetric and the proof for the Laplace mechanism follows from replacing the normal distribution with the Laplace distribution.

# B  Details for Section 5

## B.1  Proof of Proposition 11 for Randomized Response

Here we show that Proposition 11 holds using a simple mechanism. The mechanism is similar to randomized response [War65] which is used in differential privacy to privately release bits. The mechanism takes a dataset as input and randomly outputs a single bit. The output is weighted towards 0 if all entries of the dataset are 0 and towards 1 otherwise. Here we use this mechanism for the proof because the calculations and presentation are particularly clean and simple since there are only two outputs. A similar proof can be used to verify the accuracy of the estimated plots for the Laplace mechanism presented in Section 5 by calculating the exact hockey-stick divergence at, e.g., $\varepsilon = 0.25$ and $\varepsilon = 1.5$.

$$\mathcal{M}(D) = \begin{cases} b & \text{with probability } \frac{3}{4} \\ 1 - b & \text{with probability } \frac{1}{4} \end{cases}$$

where $b \in \{0, 1\}$ is 0 if all entries in $D$ are 0 and 1 otherwise.

We use the dataset $D$ that consists of all zeroes and $D'$ is obtained from $D$ by adding a single 1. We will present the proof using $\mathcal{M}_{Poisson}$, but it is the same for $\mathcal{M}_{WOR}$ since the only effect on the output distribution is whether or not the 1 is sampled in a batch. We use a sampling probability of $\gamma = 1/2$. Since the output distribution of $\mathcal{M}$ is symmetric this means that the probability for $\mathcal{M}_{Poisson}(D')$ to output either bit is $1/2 \cdot 3/4 + 1/2 \cdot 1/4 = 1/2$. The counterexample occurs when

running the mechanism for 2 iterations. There are 4 possible outcomes of the two iterations. The probability of any of these outcomes for $\mathcal{M}_{Poisson}(D')$ is $1/2 \cdot 1/2 = 1/4$. For $\mathcal{M}_{Poisson}(D)$ the probability we can find the output distribution by considering each distinct outcome

$$\Pr[\mathcal{M}_{Poisson}(D) \times \mathcal{M}_{Poisson}(D) = (0,0)] = \Pr[\mathcal{M}_{Poisson}(D) = 0] \cdot \Pr[\mathcal{M}_{Poisson}(D) = 0] = 3/4 \cdot 3/4 = 9/16$$
$$\Pr[\mathcal{M}_{Poisson}(D) \times \mathcal{M}_{Poisson}(D) = (0,1)] = \Pr[\mathcal{M}_{Poisson}(D) = 0] \cdot \Pr[\mathcal{M}_{Poisson}(D) = 1] = 3/4 \cdot 1/4 = 3/16$$
$$\Pr[\mathcal{M}_{Poisson}(D) \times \mathcal{M}_{Poisson}(D) = (1,0)] = \Pr[\mathcal{M}_{Poisson}(D) = 1] \cdot \Pr[\mathcal{M}_{Poisson}(D) = 0] = 1/4 \cdot 3/4 = 3/16$$
$$\Pr[\mathcal{M}_{Poisson}(D) \times \mathcal{M}_{Poisson}(D) = (1,1)] = \Pr[\mathcal{M}_{Poisson}(D) = 1] \cdot \Pr[\mathcal{M}_{Poisson}(D) = 1] = 1/4 \cdot 1/4 = 1/16$$

Now, we find the hockey-stick divergence in both directions for $\alpha = 4/3$ and $\alpha = 2$. We denote the two distributions for running the mechanism as $P = \mathcal{M}_{Poisson}(D) \times \mathcal{M}_{Poisson}(D)$ and $Q = \mathcal{M}_{Poisson}(D') \times \mathcal{M}_{Poisson}(D')$.

$$H_{4/3}(P\|Q) = \Pr[P = (0,0)] - 4/3 \cdot \Pr[Q = (0,0)] \qquad\qquad = 9/16 - 4/3 \cdot 1/4 = 11/48$$
$$H_{4/3}(Q\|P) = \Pr[Q \in \{(0,1),(1,0),(1,1)\}] - 4/3 \cdot \Pr[P \in \{(0,1),(1,0),(1,1)\}] \quad = 3/4 - 4/3 \cdot 7/16 = 1/6$$
$$H_2(P\|Q) = \Pr[P = (0,0)] - 2 \cdot \Pr[Q = (0,0)] \qquad\qquad = 9/16 - 2 \cdot 1/4 = 1/16$$
$$H_2(Q\|P) = \Pr[Q = (1,1)] - 2 \cdot \Pr[P = (1,1)] \qquad\qquad = 1/4 - 2 \cdot 1/16 = 1/8$$

As such, we have that $H_{4/3}(P\|Q) > H_{4/3}(Q\|P)$ and $H_2(P\|Q) < H_2(Q\|P)$.

## B.2 Details of Monte Carlo Simulation

To produce Figure 1, we leverage the PLD framework and apply Monte Carlo simulation.

By Proposition 9 and Theorem 8, the privacy curve of the composed and subsampled Laplace mechanism under add (remove) is given by $H_{e^\varepsilon}(\mathcal{M}_{Poisson}(D)^k\|\mathcal{M}_{Poisson}(D')^k)$ (vice-versa for remove) where

$$D := (0,\ldots,0) \quad D' := (0,\ldots,0,1).$$

On the other hand, a standard result (e.g. Theorem 3.5 of [GLW21]) asserts that the PLD of a composed mechanism is obtained by self-convolving the PLD of the uncomposed mechanism, namely

$$H_{e^\varepsilon}(\mathcal{M}_{Poisson}(D)^k\|\mathcal{M}_{Poisson}(D')^k) = \mathbb{E}_{Y \sim L_{\mathcal{M}_{Poisson}^k}(D\|D')}[1 - e^{\varepsilon - Y}]$$
$$= \mathbb{E}_{Y \sim L_{\mathcal{M}_{Poisson}}(D\|D')^{\oplus k}}[1 - e^{\varepsilon - Y}].$$

We estimate this expectation via sampling. We know the densities of $\mathcal{M}_{Poisson}(D) = \mathcal{N}(0, \sigma^2)$ and $\mathcal{M}_{Poisson}(D') = (1-\gamma)\mathcal{N}(0,\sigma^2) + \gamma\mathcal{N}(1,\sigma^2)$, so we can quickly sample $L_{\mathcal{M}_{Poisson}}(D\|D')$. By drawing $k$ samples and summing them, we can sample $L_{\mathcal{M}_{Poisson}}(D\|D')^{\oplus k}$ as well. Therefore, we can draw $Y_i \sim L_{\mathcal{M}_{Poisson}}(D\|D')^k$ for $1 \le i \le N$, then compute the Monte Carlo estimate

$$\frac{1}{N}\sum_{i=1}^{N}(1 - e^{\varepsilon - Y_i}).$$

As for the error, the quantity inside the expectation is bounded in $[0, 1]$, so we can apply Höffding as well as the union bound. In this case,

$$N = \left\lceil \frac{\ln(2|E|/\beta)}{2\alpha^2} \right\rceil$$

samples will suffice to ensure that the Monte Carlo estimate of $H_{e^\varepsilon}(\mathcal{M}_{Poisson}(D)\|\mathcal{M}_{Poisson}(D'))$ is accurate within $\alpha$, with probability $1 - \beta$, for all $\varepsilon \in E$ simultaneously.

For Figure 1, we chose $\alpha = 0.001$ and $\beta = 0.01$ and considered $|E| = 40$ values of $\varepsilon$, which required $N = 3,342,306$ samples. This value of $\alpha$ is small enough relative to the plot that our conclusion holds with probability 99%.

## C  Proof of Proposition 13

The proof relies mainly on the following data-processing inequality, which can also be seen as closure of privacy under post-processing.

**Lemma 15.** *Let $P$ and $Q$ be any distributions on $\mathcal{X}$ and let $\mathrm{Proc} : \mathcal{X} \to \mathcal{Y}$ be a randomized procedure. Denote by $\mathrm{Proc}\,P$ the distribution of $\mathrm{Proc}(X)$ for $X \sim P$. Then, for any $\alpha \geq 0$,*

$$H_\alpha(\mathrm{Proc}\,P || \mathrm{Proc}\,Q) \leq H_\alpha(P||Q).$$

*Proof.* For any event $E \subseteq \mathcal{Y}$,

$$
\begin{aligned}
(\mathrm{Proc}\,P)(E) - \alpha(\mathrm{Proc}\,Q)(E) &= \mathbb{E}_{\mathrm{Proc}}[\mathbb{P}_{X \sim P}(\mathrm{Proc}(X) \in E)] - \alpha\mathbb{E}_{\mathrm{Proc}}[\mathbb{P}_{X \sim Q}(\mathrm{Proc}(X) \in E)] \\
&= \mathbb{E}_{\mathrm{Proc}}[P(\mathrm{Proc}^{-1}(E))] - \alpha\mathbb{E}_{\mathrm{Proc}}[Q(\mathrm{Proc}^{-1}(E)] \\
&= \mathbb{E}_{\mathrm{Proc}}[P(\mathrm{Proc}^{-1}(E)) - \alpha Q(\mathrm{Proc}^{-1}(E)] \\
&\leq \mathbb{E}_{\mathrm{Proc}}[H_\alpha(P||Q)] \\
&= H_\alpha(P||Q)
\end{aligned}
$$

and the result holds since

$$H_\alpha(\mathrm{Proc}\,P || \mathrm{Proc}\,Q) = \sup_{E \subseteq \mathcal{Y}} (\mathrm{Proc}\,P)(E) - \alpha(\mathrm{Proc}\,Q)(E).$$

$\square$

We now prove the proposition. Our main goal is to argue that $D := (0, \ldots, 0, 1)$ and $D' := (0, \ldots, 0, -1)$ form a dominating pair of datasets for $\mathcal{M}_{Poisson}$. To that end, consider any $\sim_S$-neighbors that differ, without loss of generality, in the last entry, say $(x, a)$ and $(x, a')$. We leverage postprocessing to show that $(\mathcal{M}_{Poisson}(x, a), \mathcal{M}_{Poisson}(x, a'))$ is dominated by $(\mathcal{M}_{Poisson}(\mathbf{0}, a), \mathcal{M}_{Poisson}(\mathbf{0}, a'))$. Indeed, consider

$$\mathrm{Proc}(y) := y + \sum_{i=1}^{|\hat{x}|} \hat{x}_i$$

where $\hat{x}$ is randomly drawn from $x$ by Poisson($\gamma$)-subsampling. Now, sampling $\mathcal{M}_{Poisson}(\mathbf{0}, a)$ is equivalent to drawing $\hat{a}$ from the singleton dataset $(a)$ via Poisson($\gamma$) and returning a sample from $\mathcal{N}(\sum_{i=1}^{|\hat{a}|} \hat{a}_i, \sigma^2)$. Since the normal distribution satisfies $\mathcal{N}(a, \sigma^2) + b = \mathcal{N}(a + b, \sigma^2)$, sampling $\mathrm{Proc}(\mathcal{M}_{Poisson}(\mathbf{0}, a))$ is equivalent to sampling

$$\mathcal{N}\left(\sum_{i=1}^{|\hat{x}|} \hat{x}_i + \sum_{i=1}^{|\hat{a}|} \hat{a}_i, \sigma^2\right)$$

where $\hat{x}$ is Poisson($\gamma$)-subsampled from $x$ and $\hat{a}$ is Poisson($\gamma$)-subsampled from $(a)$. But, by independence, $(\hat{x}, \hat{a})$ is a Poisson($\gamma$)-subsample drawn from $(x, a)$, so, in conclusion, $\mathrm{Proc}(\mathcal{M}_{Poisson}(\mathbf{0}, a)) = \mathcal{M}_{Poisson}(x, a)$. By an analogous argument, we have that $\mathrm{Proc}(\mathcal{M}_{Poisson}(\mathbf{0}, a')) = \mathcal{M}_{Poisson}(x, a')$ and hence

$$
\begin{aligned}
H_\alpha(\mathcal{M}_{Poisson}(x, a) || \mathcal{M}_{Poisson}(x, a')) &= H_\alpha(\mathrm{Proc}(\mathcal{M}_{Poisson}(\mathbf{0}, a)) || \mathrm{Proc}(\mathcal{M}_{Poisson}(\mathbf{0}, a'))) \\
&\leq H_\alpha(\mathcal{M}_{Poisson}(\mathbf{0}, a) || \mathcal{M}_{Poisson}(\mathbf{0}, a')) \quad \text{(Lemma 15)} \\
&\leq H_\alpha(\mathcal{M}_{Poisson}(\mathbf{0}, 1) || \mathcal{M}_{Poisson}(\mathbf{0}, -1)).
\end{aligned}
$$

## D  Constructing a Dominating Pair of Distributions for the Gaussian Mechanism

In this section we consider the problem of computing privacy curves for the Gaussian mechanism under $\sim_S$ when sampling without replacement. As shown in Section 7 computing tight parameters is challenging in this setting because we do not know which datasets result in the largest hockey-stick divergence. However, we can still compute an upper bound on the privacy curve using a dominating pair of distributions.

We modified the implementation of the algorithm introduced by [DGK$^+$22] in the Google DP library to construct the PLDs (Privacy Loss Distribution object). The algorithm constructs an approximation of the PLD from the hockey-stick divergence between the pair of distributions at a range of values for $\varepsilon$. From Theorem 14 we know that the direction of the pair of distributions yielding the largest hockey-stick divergence for the mechanism of a single iteration differs for $\alpha$ below and above 1. We construct a new PLD by combining the two directions at $\alpha = 1$ or $\varepsilon = 0$.

See the left-side plot of Figure 4 for a visualization of how our construction uses the point-wise maximum of the hockey-stick divergence for a single iteration. This construction represents a dominating pair of distributions and as such it is sufficient to find a dominating pair of distributions for the composed mechanism using self-composition by Theorem 8.

The right-side plot of Figure 4 shows the privacy curve obtained from self-composing the PLD for the dominating pair of distributions with parameters $\sigma = 4$, $\gamma = 0.05$, and 1000 iterations. The blue line is the privacy curve under $\sim_R$ and also serves as a lower bound for the true privacy curve. Note that the orange line would also be the privacy curve achieved by this technique under the add/remove relation if we did not consider the add and remove relations separately.

The gap between the upper and lower bound motivates future work for understanding the worst-case datasets. Similar to the add/remove case we conjecture that the subsampled Gaussian mechanism behaves well under composition. Specifically, we conjecture that the privacy curve of the composed subsampled Gaussian mechanism under $\sim_S$ matches the curve under $\sim_R$ for $\varepsilon \geq 0$. It seems likely that this is the case if Conjecture 12 holds. However, if Conjecture 12 does not hold the above statement also does not hold.

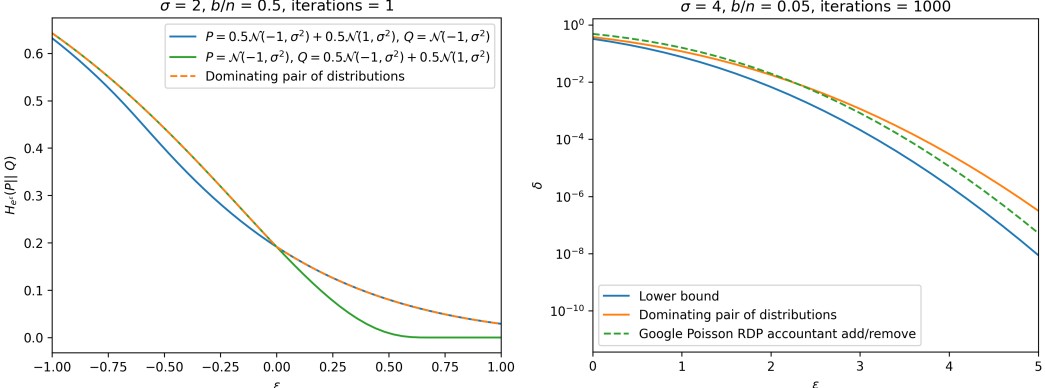

Figure 4: Hockey-stick divergence for the Gaussian mechanism under substitution when sampling without replacement using a dominating pair of distributions. The dominating pair of distributions is constructed using a point-wise maximum of the privacy curve for a single iteration as seen in the left plot. The right plot compares the privacy curve from self-composing the dominating pair of distributions with a lower bound obtained from self-composing the PLD that corresponds to the blue line in the left plot. The dotted line for the RDP accountant is used for reference of scale. The difference between the blue and the dotted line corresponds to the difference between using the PLD and RDP accountants for Poisson subsampling under add/remove.

