# OpenReview forum: "Avoiding Pitfalls for Privacy Accounting of Subsampled Mechanisms under Composition"
_NeurIPS.cc/2024/Conference — Submitted to NeurIPS 2024_

### Official Review · Reviewer_DUtv · 2024-07-05

**Soundness:** 4
**Presentation:** 4
**Contribution:** 2
**Rating:** 5
**Confidence:** 5

**Summary:**

The authors present two potential sources of error which can arise when composing sub-sampled DP mechanisms. On one hand, they discuss cases in which the composition of worst-case datasets does not yield the expected result, on the other hand, they disambiguate guarantees for mechanisms with Poisson sampling vs. sampling WOR.

**Strengths:**

I appreciate that this paper points out some of the subtleties which arise regarding the distinctions between worst-case databases and dominating pairs and regarding the correct accounting of specific sampling schemes (Poisson/WOR). These subtleties can pass unnoticed, and lead to errors which compromise the privacy of individuals.

**Weaknesses:**

There is nothing particularly wrong with the paper. The facts stated are valid, and they are interesting, especially since they point out potential sources of confusion. However, none of this is surprising or even particularly novel. Most of this information is implied by earlier work (Zhu et al. in particular), and some of the facts stated here would be better suited as GitHub issues on the relevant accountants, followed by a technical report at a venue like TPDP or Journal of Privacy and Confidentiality. That is to say: I am not against this paper in general, but this is not a NEURIPS paper to me. It is a highly specialised piece of technical writing with a very narrow scope, and is likely to be of interest only to a very small community. I would recommend the authors to submit it to a venue which is better suited to its content.

**Questions:**

Some suggestions for improvement:

- Definition 1: The "only if" does not hold here. For example, the mechanism also satisfies epsilon/delta-DP if it satisfies epsilon/delta probabilistic DP, i.e. the PLRV is bounded with probability at least 1-delta, but not the other way around.
- Definition 3: The hockey-stick divergence is asymmetric in general. A more appropriate terminology would be "of P from Q"
- There are some measure-theoretic constructs used, but there are no assumptions stated about absolute continuity (think (0, delta)-DP) or about whether densities exist (Definition 5), and the notation in Definition 5 is a bit uncommon. A nitpick: the "d" is a differential operator, and it's recommended to write it as $\mathrm{d}$
- There is a subtlety about Poisson sampling in Definition 6: Treating the **expected batch size** as public is fine, but the footnote just says "batch size". The **actual** batch size (the realised one), should still be kept secret right?
- 160: Referring to $D \sim D'$ as a dominating pair of datasets is a recipe for confusion in my opinion. The established terminology is to refer to dominating pairs of measures (or distributions). Perhaps state that the distributions under D, D' are dominating pairs, but not the databases themselves. After all, the whole point of even considering dominating pairs is (allegedly) to not have to think in terms of databases.
- Perhaps use "databases" consistently rather than mixed with "datasets"
- In 244: Your paper is aimed at an audience who are not DP theorists. I think just stating that "they converge to a Gaussian distribution" does not really inform this audience about why the crossing vanishes. You'd probably have to explain that the privacy profiles become pointwise equal because the PLDs now become symmetric about the expectations.
- In Figure 2, just because the mechanisms satisfy the same delta and epsilon, this does not make them identical or even comparable in general (they are only "identical" if they have equal privacy curves or trade-off functions). The figure seems to rely in a sense on the mechanisms being otherwise identical. Is there a limitation which needs addressing here?
- The Gopi accountant is known to break for low delta. Did you encounter any problems for delta 1e-7? Is there any chance the results could be spurious for low delta values?

**Limitations:**

The discussion on limitations is a bit lacking in my opinion. The authors state (in the checklist) that the "main limitation is expressed in Conjecture 12". Not being able to find a counterexample for a proposition is not really what one understands under the term "limitation" of a work. In particular, the supposed "limitation" is --by the authors' own admission-- easily resolved by just running the accountant on the two curves separately and taking the supremum. I would have much preferred an experimental section where the consequences of the pitfalls stated in the work are actually shown to affect the real-world use of DP-SGD or other mechanisms, and/or to see that specific privacy threats are practically enabled by overlooking these subtleties (e.g. through auditing).

---

> ### Author Rebuttal · Authors · 2024-08-06
>
> We appreciate the reviewer agreeing that the findings are interesting.
> We respectfully disagree that the findings may be too niche or specialized, or too "narrow [in] scope" for highlighting at a conference like NeurIPS.
> As just one example, the concurrent work ``How Private are DP-SGD Implementations?'' by Chua et al. focuses precisely on issues related to discrepancies between DP-SGD implementations and accounting techniques.
> Their work could be similarly criticized as being narrow in scope or niche.
> However, this paper was not only accepted to ICML 2024, but awarded an oral presentation (only 6\% of accepted papers), recognizing it as one of the papers most deserving of highlighting to the community.
> Our work is thematically quite similar in focus to theirs.
> And while the reviewer claims that the results are not surprising or novel, we again respectfully disagree.
> Indeed, similar to almost all work in the field, we build upon prior contributions.
> It appears that the reviewer may have substantial technical expertise with prior contributions, and thus some of the arguments we employ may seem more natural to them.
>
> In summary, we believe the reviewer (at a minimum) agrees with us that the paper highlights and demonstrates interesting, important, and practically-relevant phenomena that are still a common pitfall for practitioners.
> Is this not sufficient for a NeurIPS paper?
> Below, we address and clarify on many of the interesting and insightful questions raised by the reviewer.
> It seems like the primary negativity in the reviewer's evaluation is based on whether this paper "feels" like a NeurIPS paper.
> We hope the reviewer reconsiders their feelings here, given the fact that NeurIPS is a "big tent" community and in light of the recognition for the thematically-similar paper of Chua et al.
>
> Furthermore, thank you for your suggestions. We will make edits to improve clarity. Below we address a couple of specific points in more detail.
>
> > In Figure 2, just because the mechanisms satisfy the same delta and epsilon, this does not make them identical or even comparable in general (they are only "identical" if they have equal privacy curves or trade-off functions). The figure seems to rely in a sense on the mechanisms being otherwise identical. Is there a limitation which needs addressing here?
>
> Table 1 does address this limitation in the sense that it demonstrates the same discrepancy even as $\delta$ is permitted to vary by several orders of magnitude.
>
> > The Gopi accountant is known to break for low delta. Did you encounter any problems for delta 1e-7? Is there any chance the results could be spurious for low delta values?
>
> For Figure 2 we used the Opacus accountant, which to our knowledge does not have this issue. In any case, the reported issues for the Gopi accountant seem to emerge for much smaller values (e.g. $10^{-14}$).
> Still, the reviewer is correct that we cannot rule out the existence of a bug in the Opacus implementation. In our research we ran similar experiments with different accountants including implementations based on RDP and found a similar gap between Poisson and WOR.
> We will add a short note to the paper.
>
> > In particular, the supposed "limitation" is --by the authors' own admission-- easily resolved by just running the accountant on the two curves separately and taking the supremum.
>
> It is true that the issue can be easily resolved in the context of the add/remove neighboring relation but the problem becomes significantly murkier under the substitution relation (as in Section 7). In that case, the number of worst-case datasets scales with the number of compositions. On the other hand, an accountant that "smooths over" worst-case datasets may be unnecessarily lossy (as in Section D). Therefore, there is an accuracy and computational motivation to settle the issue of worst-case datasets under the substitution relation. Resolving the conjecture for the add/remove relation will lead to a solution for the substitution relation. Our paper is unclear on this point and we will rephrase to emphasize it.

---

> > ### Comment · Reviewer_DUtv · 2024-08-08
> > **Thank you and response to rebuttal**
> >
> > Thank you for addressing the points raised in my review. I consider my suggestions for clarity etc., adequately addressed by the comments, and I'm confident that you have the technical expertise in DP to incorporate (or not) the other comments which I made and which were not specifically addressed point to point in your rebuttal. One final thought:
> >
> > > We respectfully disagree that the findings may be too niche or specialized, or too "narrow [in] scope" for highlighting at a conference like NeurIPS. As just one example, the concurrent work ``How Private are DP-SGD Implementations?'' by Chua et al. focuses precisely on issues related to discrepancies between DP-SGD implementations and accounting techniques. Their work could be similarly criticized as being narrow in scope or niche. However, this paper was not only accepted to ICML 2024, but awarded an oral presentation (only 6% of accepted papers), recognizing it as one of the papers most deserving of highlighting to the community. Our work is thematically quite similar in focus to theirs. And while the reviewer claims that the results are not surprising or novel, we again respectfully disagree. Indeed, similar to almost all work in the field, we build upon prior contributions. It appears that the reviewer may have substantial technical expertise with prior contributions, and thus some of the arguments we employ may seem more natural to them.
> >
> > I agree with you that the discussion about scope and novelty has a strong subjective element, and I'm not going to hold this point against you. In fact, I'm willing to give you the benefit of the doubt and increase my score. However, I feel like I have to push back a bit against your argumentation using the Chua et al. paper. The paper (largely) does away with an age-old debate: "If we use the non-private SGD version of iterating through minibatches one by one, what happens to privacy guarantees?" Opacus, during its entire 0.XX release cycle had a note to the effect of "this type of minibatch sampling can be a good approximation of Poisson sampling" (I'm paraphrasing), and I'm fairly sure that many users and even people with some knowledge of DP came away with the notion that not using Poisson sampling is in some way OK. In other words: the Chua et al. paper takes big strides towards cleaning up a **major** misunderstanding, and I thus understand why it was awarded an oral at ICML. While indeed, thematically, your work is in a similar flavour, I am hard-pressed to view the issue you are tackling here as having the same impact or scope. I will increase my score as I recognise that the topic of your work also has importance (and because "in DP, details matter"), but I maintain that the impact and scope of your results, while interesting and valid, are not on par with some of the most important results in the field.

---

### Official Review · Reviewer_Tsvf · 2024-07-08

**Soundness:** 3
**Presentation:** 2
**Contribution:** 3
**Rating:** 6
**Confidence:** 2

**Summary:**

This paper examines the discrepancies between privacy accounting methods and their implementations, highlighting several cases where these mismatches lead to incorrect results. Specifically, it compares the noise requirements for achieving privacy guarantees under Poisson sampling versus sampling without replacement, and explores the limitations of worst-case dataset assumptions in subsampled mechanisms. Additionally, the authors address challenges in computing tight differential privacy (DP) bounds under the substitution relation of neighboring datasets.

**Strengths:**

- The paper addresses an important and timely topic in the field of differential privacy regarding privacy accounting.

- The findings have strong practical implications, potentially preventing unintended privacy breaches.

- The authors' message is well articulated, promoting better practices among DP practitioners.

- Despite critiquing existing methods, the authors maintain a respectful and constructive tone.

**Weaknesses:**

- The different messages of the paper may be convoluted sometimes, which makes the paper hard to follow.

- No viable technical solutions are provided for the identified issues, which might be a difficult research problem.

**Questions:**

- In the equation between lines 133 and 134 (and in the rest of the article), where did the maximum with $0$ go ? When I try to recover expression on the right-hand side, I obtain $E_Y \max(0, 1-e^{\epsilon - Y})$.

**Limitations:**

The authors discuss the limitations of their work.

---

> ### Author Rebuttal · Authors · 2024-08-06
>
> > No viable technical solutions are provided for the identified issues, which might be a difficult research problem.
>
> As the reviewer suggests, this is indeed a very difficult research direction, which we have invested substantial time and effort into.
> The main intention of our work is to draw theorist and practitioner attention to these issues, and clearly and crisply state reasonable technical conjectures for the community to coalesce around and attack.
> Traditionally, having clear and well-defined technical problems and conjectures has spurred mathematical advancements and progress.
> We consider these issues to be of great enough practical importance that we urge the community to work on these problems with us in parallel.
>
> > In the equation between lines 133 and 134 (and in the rest of the article), where did the maximum with 0 go?
>
> This is a typo. Thank you for pointing it out!

---

> > ### Comment · Reviewer_Tsvf · 2024-08-10
> >
> > Thank you for addressing the points raised in my review. As pointed out by the authors, correctly accounting privacy with algorithms such as DPSGD is a crucial research direction. Even though the authors do not provide technical solutions to the problems shown in the article, their identification is still meaningful. If the typo that I pointed out has no impact on the conclusion of the authors (i.e. if it didn't introduce errors later in the paper), I will improve my rating from 5 to 6.

---

### Official Review · Reviewer_NpBG · 2024-07-17

**Soundness:** 3
**Presentation:** 3
**Contribution:** 3
**Rating:** 5
**Confidence:** 3

**Summary:**

The two main contributions of the paper is as follows:

the privacy guarantee of composition of subsampled mechanism may not be defined by worst-case dataset(s) for the underlying mechanism
Poisson subsampling and sampling without replacement may not have similar privacy guarantee.

**Strengths:**

The paper studies a very important problem of composition of subsampled privacy mechanism. There has been a lot of work in the recent past that performs a tight privacy accounting. This work is in the line of these works. These accounting results are used in deployment as well to show how much privacy loss has happened during training when using a prescribed noise scale. Based on these bounds, the training is stopped once we have expired the privacy budget. In this regard, their second result is very important because we definitely use subsampling without replacement in DP-SGD.

**Weaknesses:**

There are some typos, and the result for the gap is shown empirically. I have to state that I have not seen the Appendix so if the authors have a provable guarantee for this gap in the Appendix, please point me that. To me, the selling point of the paper is this result and it should be placed front and center. Most of the results that are given in the form of propositions and lemma are from previous works.

**Questions:**

Is there a provable gap between the composition of WoR and Poisson subsampling?

**Limitations:**

Mostly seem like an empirical study of the composition result.

---

> ### Author Rebuttal · Authors · 2024-08-06
>
> > There are some typos
>
> If the reviewer could point out specific typos that they noticed that would be highly appreciated.
>
> > ...the result for the gap is shown empirically. I have to state that I have not seen the Appendix so if the authors have a provable guarantee for this gap in the Appendix, please point me that. To me, the selling point of the paper is this result and it should be placed front and center. Most of the results that are given in the form of propositions and lemma are from previous works.
>
> We show the result for the gap both empirically and theoretically.
> As a convention, we present results from previous work as theorems and our results as lemmas and propositions. We will make this convention explicit in the paper.
> The proofs of all propositions are in the Appendix.
> Proposition 9 shows that noise with twice the magnitude is required under WOR when compared to Poisson.
> The proof of Proposition 9 is in Appendix A.
>
> Note that this gap does not follow from the result of Zhu et al.
> They give dominating pairs of distributions for upper bounding the privacy parameters.
> We show that the bounds are tight for both Poisson and WOR in the case of the subsampled Laplace/Gaussian.
> It is likely not very surprising to privacy accounting experts that the distributions are tight for DP-SGD, but it is not true for all mechanisms.
>
> > Is there a provable gap between the composition of WoR and Poisson subsampling?
>
> Yes, please again refer to Appendix A for the details.

---

> > ### Comment · Area_Chair_YANG · 2024-08-13
> > **Quick reminder to respond to the rebuttal**
> >
> > Dear Reviewer NpBG,
> >
> > As we approach the end of the discussion period with the authors, I would really like to hear your thoughts on their rebuttal. I would really appreciate it if you could engage with them before the close of discussions (Aug 13th, 11:59pm AoE).
> >
> > Thanks,
> > AC

---

### Official Review · Reviewer_qk3S · 2024-07-19

**Soundness:** 2
**Presentation:** 3
**Contribution:** 2
**Rating:** 3
**Confidence:** 4

**Summary:**

This paper studies the notion sampling with replacement for differential privacy. Most of the literature on machine learning with differential privacy benefits from privacy amplification by poisson sampling in the privacy analysis. However, when implementing the mechanisms, engineers ofter use the sub-sampling with replacement as a substitude for poisson sampling, mainly due to efficiency issues. This paper studies the gap between these two settings. Their main contributions are as follows:

 - Identifying the Problem with DP-SGD Implementations: The authors highlight a critical issue with implementations of (DP-SGD). They argue that many implementations incorrectly assume that Poisson sampling and batch-sampling yield similar privacy guarantees, which is not necessarily true.

- Gap between Batch-Sampling and Poisson Sampling: The paper demonstrates a significant privacy gap between these two sampling methods. They provide an example showing that for certain hyperparameters, Poisson subsampling can result in an ϵ≈1, whereas batch-sampling without replacement can result in an ϵ>10. This discrepancy is critical for privacy accounting in DP-SGD.  Authors compare the privacy guarantees of Poisson subsampling and batch-sampling. They show that the privacy guarantees can differ significantly depending on the sampling technique used. Their analysis reveals that the method of sampling batches (Poisson vs. fixed-size) significantly impacts the resulting privacy guarantees, cautioning against the interchangeable use of different sampling techniques in privacy analysis.

**Strengths:**

- Identifying an important problem with implementation of DP-SGD

**Weaknesses:**

- I have some concerns about the correctness of the results.

- There is not much technical novelty.

**Questions:**

- You introduce the notion of dominating pairs of distributions, but then you talk about dominating datasets. You need to clarify the relation. Specifically in Proposition 9, I don't understand what a dominating dataset means.

- Proposition 9 looks incorrect to me. In case of b=n and \lambda=1, the dominating pairs should collide, but this proposition suggests otherwise.

- You say: "Crucially, Proposition 9 implies that under the add and remove relations, we must add noise with twice the magnitude when sampling without replacement compared to Poisson subsampling!". This doesn't sound correct. Again, if you set the sampling rate to 1, then the two mechanisms collide. Am I missing something?

- For the results of section 6, are you using the worst-case datasets of Proposition 9 to calculate the privacy curve?

**Limitations:**

Yes

---

> ### Author Rebuttal · Authors · 2024-08-06
>
> There are no issues with the correctness of the results questioned by the reviewer. We discuss the technical details of Proposition 9 below.
>
> > You introduce the notion of dominating pairs of distributions, but then you talk about dominating datasets. You need to clarify the relation. Specifically in Proposition 9, I don't understand what a dominating dataset means.
>
> By dominating datasets we mean a pair of datasets that induce a dominating pair of distributions when the mechanism is applied to them. The distinction is important for our results. We have been unclear about this point so thank you for mentioning it. We will update the text to clarify.
>
> > Proposition 9 looks incorrect to me. In case of $b=n$ and $\lambda=1$, the dominating pairs should collide, but this proposition suggests otherwise.
>
> We assume that $\lambda$ refers to the Poisson subsampling rate, which we call $\gamma$. The dominating pairs will not collide in this case due to the fact that sampling without replacement involves a fixed batch size and we are considering the add/remove relation. Therefore the sampling rate differs between neighboring datasets. Taking $n$ to be the size of the larger dataset, the largest value we can choose for $b$ is $n-1$, the size of the smaller dataset. Taking $b = n-1$ leads to a rate for sampling without replacement of $(n-1)/n$, so for a direct comparison we should consider $\gamma = (n-1)/n$ for Poisson as well instead of $\gamma = 1$.
>
> Note that it is not an artifact of the choice that $n$ refers to the size of the larger dataset. If we refer to the size of the smaller dataset as $n$ such that $b = n$ is valid we see a similar difference. But the comparison is not as clean with this choice because we would use $2n/(n+1)$ times as much noise.
>
> > You say: "Crucially, Proposition 9 implies that under the add and remove relations, we must add noise with twice the magnitude when sampling without replacement compared to Poisson subsampling!". This doesn't sound correct. Again, if you set the sampling rate to 1, then the two mechanisms collide. Am I missing something?
>
> For the settings of $b$ and $\gamma$ we just described, the dominating pair we obtain for Poisson subsampling is $\mathcal{N}(0, \sigma^2)$ vs. $\frac{1}{n}\mathcal{N}(0, \sigma^2) + \frac{n-1}{n}\mathcal{N}(1, \sigma^2)$. For sampling without replacement we get $\mathcal{N}(-b, \sigma^2)$ vs. $\frac{1}{n}\mathcal{N}(-b, \sigma^2) + \frac{n-1}{n}\mathcal{N}(-(b - 1) + 1, \sigma^2)$. Due to the translational properties of the normal distribution, this is equivalent to $\mathcal{N}(0, \sigma^2)$ vs. $\frac{1}{n}\mathcal{N}(0, \sigma^2) + \frac{n-1}{n}\mathcal{N}(2, \sigma^2)$. Thus we need twice as much noise under WOR.
>
> > For the results of section 6, are you using the worst-case datasets of Proposition 9 to calculate the privacy curve?
>
> Yes.

---

> > ### Comment · Area_Chair_YANG · 2024-08-13
> > **Quick reminder to respond to the rebuttal**
> >
> > Dear Reviewer qk3S,
> >
> > As we approach the end of the discussion period with the authors, I would really like to hear your thoughts on their rebuttal. I would really appreciate it if you could engage with them before the close of discussions (Aug 13th, 11:59pm AoE).
> >
> > Thanks,
> > AC

---

> > ### Comment · Reviewer_qk3S · 2024-08-14
> > **Thank you for the rebuttal**
> >
> > I'm not sure if I understand the notion of dominating databases. What prevents you from choosing the neighboring datasets to be
> >
> > $$D=(0,...,0,0) ~~ {and} ~~ D'=(0,...,0,100)?$$
> >
> > This clearly incurs higher hockey-stick divergence compared to $$D=(0,...,0,0) ~~ {and} ~~ D'=(0,...,0,1).$$
> >
> > So I think the notion of dominating databases should be normalized by the sensitivity.
> >
> > Now I think you are actually making this mistake in your proposition 9. Your neighboring datasets for the case of poisson sampling has sensitivity of 1.0, while your neighboring dataset for the case of sampling with replacement has sensitivity 2.0. This is why you can show that you need double the noise! But this is meaningless.
> >
> > The example I gave is actually a good way to see why your result isn't meaningful. Even if we consider the sampling rate to be (n-1)/n (I don't know why you are doing this but it's ok), the two mechanisms will collide as n approaches to infinity.

---

> ### Author Response · Authors · 2024-08-14
>
> > What prevents you from choosing the neighboring datasets to be $D=(0,...,0,0) ~~ {and} ~~ D’=(0,...,0,100)$?
>
> Please see the paragraph beginning at l.90. In our work we consider mechanisms over a bounded domain. We restrict to the domain [-1, 1], without loss of generality. Without a bounded domain (or, equivalently, some sort of clipping), as the instance the reviewer highlights exemplifies, the sensitivity may be arbitrarily large and no non-trivial results are at all possible for the Laplace or Gaussian Mechanism. Note, in particular, that the Subsampled Gaussian Mechanism, which forms the basis of DPSGD, employs clipping for precisely this reason.
>
> > Now I think you are actually making this mistake in your proposition 9. Your neighboring datasets for the case of poisson sampling has sensitivity of 1.0, while your neighboring dataset for the case of sampling with replacement has sensitivity 2.0
>
> First, we maintain that Proposition 9 is correct. The proof (in Appendix A) is rather succinct (~half a page), and if the reviewer can identify any specific technical issue in the proof, we would be willing to reconsider our position.
>
> That said, the reviewer has highlighted one of the most surprising and counterintuitive results in our work! Indeed, it certainly "feels" like the datasets D1 = (-1, ..., -1) and D1' = (-1, ..., -1, 1) ought to be "worse" than D2 = (0, ..., 0) and D2' = (0, ..., 0, 1). Indeed, this is correct for WOR: as the reviewer points out, including the 1 will make a bigger difference if it displaces a -1 (when the difference is 2) rather than if it displaces a 0 (when the difference is 1).
>
> However, quite surprisingly, the same does not hold for Poisson! We give an informal argument of why that is here. In case 1 (with -1's), we could achieve a particular sum because the underlying dataset is D1 and a -1 is not sampled (which increases the sum by 1), or because the underlying dataset is D1' and the +1 is sampled (which increases the sum by 1). The same confusion can not occur for D2 and D2', since for D2 the sum is always fixed to be 0, and in D2' the sum is 1 with probability $\gamma$. That is, there is no event under D2 that "looks like" a 1 has been included, whereas there is for D1 (where a -1 is excluded).
>
> Of course, the discussion above is not a precise argument. However, the rigorous technical details are present in Appendix A of our paper and appeal to Theorem 10.
>
> > The example I gave is actually a good way to see why your result isn’t meaningful. Even if we consider the sampling rate to be (n-1)/n (I don’t know why you are doing this but it’s ok), the two mechanisms will collide as n approaches to infinity.
>
> We remind that we considered the sampling rate of (n-1)/n because this is the closest possible case to the $\gamma = 1$ case suggested by the reviewer (due to the discrepant size of the datasets). We also comment that the two mechanisms would in fact not collide, either with the same pair of datasets (though this can be shown directly via a sophisticated argument regarding the rather complex mixtures induced by the Poisson sampling on the -1's dataset, the easiest way to see it is by observing that it would contradict our proof which does not have any issues) or the pairs of datasets that we define in Proposition 9 (as demonstrated in our initial rebuttal, please refer to it above).

---

### Decision · Program_Chairs · 2024-09-25

**Decision:**

Reject

**Comment:**

The submission identifies critical issues with differential privacy (DP) accounting methods, specifically highlighting discrepancies between Poisson subsampling and sampling without replacement and problems in worst-case dataset assumptions (e.g., the worst-case dataset shift for a single composition is not the worst-case under multiple compositions). Reviewers appreciated the relevance of these issues for DP practitioners. The message is clear: DP accounting must be done with care, and naive implementations may fail by considering wrong worst-case datasets and amplification via subsampling. However, reviewers DUtv and Tsvf noted that while the paper successfully points out these challenges, it falls short of offering concrete solutions. DUtv specifically mentioned that this paper would be more appropriate for a niche venue focused on DP, rather than NeurIPS, suggesting a workshop setting might be better suited for its contributions. Furthermore, DUtv highlighted that the comparison with Chua et al.'s work seemed misplaced, noting that the impact and scope of this paper do not quite match the broader significance of Chua et al.'s findings. Reviewer qk3S raised concerns about the corretness of Proposition 9. Other reviewers did not support the issue raised by reviewer qk3S, and this concern was downplayed in the discussion and in this meta-review. Reviewer NpBG did not engage with the rebuttal, so their review is given less weight.

The main reason to reject this paper is its lack of actionable solutions for the identified privacy accounting issues (DUtv and Tsvf), making its contributions less compelling for a broad audience. While differential privacy is absolutely important and of significant interest to NeurIPS and the wider machine learning community, this paper does not provide the kind of applicable solutions needed to advance the field. As pointed out by several reviewers, identifying problems in accounting schemes is relevant for the DP community. However, going beyond raising issues and concretely developing possible solutions and clear recommendations would have been more meaningful for a broader ML audience. The reader is left wondering what are the concrete and actionable steps to accurately account for self-composition (especially when dominating pairs of PLRV are difficult to compute), and to identify the true DP guarantee from subsampling given that it is seemingly so easy to err. While some hints of solutions are dispersed throughout the paper, they are not presented in a clear and actionable way. The sentiment was shared by reviewers both during the review and the discussion phase.

I encourage the authors to continue working in this direction and, if resubmitting to an ML conference with a broad audience, consider developing actionable solutions to the important technical insights they outline in their work. The message that accounting should be done with care -- and implementations can easily be flawed -- is of interest to the DP community. However, for a broader audience, actual solutions would have been more impactful. How should we fix these issues? The two lasting recommendations in the paper, namely that practitioners must disclose privacy accounting hyperparameters for reproducibility and re-run privacy accounting when comparing different DP-SGD mechanisms, were found insufficient by the reviewers.